# Selection of Soybean and Cowpea Cultivars with Superior Performance under Drought Using Growth and Biochemical Aspects

**DOI:** 10.3390/plants12173134

**Published:** 2023-08-31

**Authors:** Rafael de Souza Miranda, Bruno Sousa Figueiredo da Fonseca, Davielson Silva Pinho, Jennyfer Yara Nunes Batista, Ramilos Rodrigues de Brito, Everaldo Moreira da Silva, Wesley Santos Ferreira, José Hélio Costa, Marcos dos Santos Lopes, Renan Henrique Beserra de Sousa, Larissa Fonseca Neves, José Antônio Freitas Penha, Amanda Soares Santos, Juliana Joice Pereira Lima, Stelamaris de Oliveira Paula-Marinho, Francisco de Alcântara Neto, Évelyn Silva de Aguiar, Clesivan Pereira dos Santos, Enéas Gomes-Filho

**Affiliations:** 1Plant Science Department, Federal University of Piauí, Teresina 64049-550, Piauí, Brazil; fneto@ufpi.edu.br; 2Postgraduate Program in Agricultural Sciences, Campus Professora Cinobelina Elvas, Federal University of Piauí, Bom Jesus 64900-000, Piauí, Brazil; ramilos@hotmail.com (R.R.d.B.); renanbiologiabomjesus@gmail.com (R.H.B.d.S.); amandasantosagro@gmail.com (A.S.S.); stelamarisop@live.com (S.d.O.P.-M.); 3Agronomic Engineering Course, Campus Professora Cinobelina Elvas, Federal University of Piauí, Bom Jesus 64900-000, Piauí, Brazil; brunofigueiredo91@ufpi.edu.br (B.S.F.d.F.); davielson5@gmail.com (D.S.P.); jennyferyaranb@gmail.com (J.Y.N.B.); everaldo@ufpi.edu.br (E.M.d.S.); wsferreira18@hotmail.com (W.S.F.); marcossantos319@gmail.com (M.d.S.L.); larissafonseca034@gmail.com (L.F.N.); joseafpenha@outlook.com (J.A.F.P.); julianalima@ufpi.edu.br (J.J.P.L.); 4Department of Biochemistry and Molecular Biology, Federal University of Ceará, Fortaleza 60451-970, Ceará, Brazil; helio.costa@ufc.br (J.H.C.); egomesf@ufc.br (E.G.-F.); 5Postgraduate Program in Environmental Sciences, Center of Sciences of Chapadinha, Federal University of Maranhão, Chapadinha 65500-000, Maranhão, Brazil; evellynas@outlook.com (É.S.d.A.); clesivan.pereira@ufma.br (C.P.d.S.)

**Keywords:** *Glycine max*, *Vigna unguiculata* L. Walp, drought tolerance, water deficit, leguminous crops

## Abstract

Identifying cultivars of leguminous crops exhibiting drought resistance has become crucial in addressing water scarcity issues. This investigative study aimed to select soybean and cowpea cultivars with enhanced potential to grow under water restriction during the vegetative stage. Two parallel trials were conducted using seven soybean (AS3810IPRO, M8644IPRO, TMG1180RR, NS 8338IPRO, BMX81I81IPRO, M8808IPRO, and BÔNUS8579IPRO) and cowpea cultivars (Aracê, Novaera, Pajeú, Pitiúba, Tumucumaque, TVU, and Xique-xique) under four water levels (75, 60, 45, and 30% field capacity—FC) over 21 days. Growth, water content, membrane damage, photosynthetic pigments, organic compounds, and proline levels were analyzed. Drought stress significantly impacted the growth of both crops, particularly at 45 and 30% FC for soybean and 60 and 45% FC for cowpea plants. The BÔNUS8579IPRO and TMG1180RR soybean cultivars demonstrated the highest performance under drought, a response attributed to increased amino acids and proline contents, which likely help to mitigate membrane damage. For cowpea, the superior performance of the drought-stressed Xique-xique cultivar was associated with the maintenance of water content and elevated photosynthetic pigments, which contributed to the preservation of the photosynthetic efficiency and carbohydrate levels. Our findings clearly indicate promising leguminous cultivars that grow under water restriction, serving as viable alternatives for cultivating in water-limited environments.

## 1. Introduction

Water scarcity has emerged as a pressing global concern, and several regions have confronted a critical shortage of water due to the combined effects of population growth, rapid urbanization, and climate change. These problems are directly reflected in detrimental consequences for agriculture, industry, and people’s health. The scarcity of water resources presents not only challenges in fulfilling fundamental human requirements but also puts ecosystems and biodiversity at risk [1,2].

In face of populational growth in recent years, leguminous crops constitute an important solution to ensure food and nutritional security, with a key socioeconomic role [2,3]. Among them, soybean (*Glycine max* L. Merrill) and cowpea (*Vigna unguiculata* L. Walp) stand out as a crucial component for humans and ruminant animals. Leguminous crops are also associated with the fixation of biological nitrogen due to their abilities in mutualistic association, thus reducing costs of industrially manufactured chemical fertilizers in supplying nitrogen to plants.

Soybean is among the most explored crops worldwide, and Brazil has become the world’s leading producer of grains, followed by the United States and China [4,5]. Nevertheless, the drought episodes are projected to promote severe decreases from 21.8 to 40% in soybean yields around the world [6,7,8]. On the other hand, the global production of cowpea reached around 9.6 million tons in 2021, with highlights in African countries [9,10]. Moreover, Brazil is among the largest producers with one million hectares of land, despite its production mainly being concentrated in the Northeast region where the drought and soil quality are abiotic factors limiting cowpea’s ability to grow satisfactorily, especially during the pod filling stage, resulting in losses above 30% [4,11,12].

In plants, several processes are disturbed by a water deficit, including morphological, physiological, and biochemical alterations, as well as modulation in gene expression. Under water restriction, plants tend to promote stomatal closure in order to avoid water loss through transpiration, also decreasing CO_2_ availability for chloroplasts, which impairs the net photosynthesis and carbohydrate biosynthesis [13,14,15]. Consequently, the energy excess in electron transport chain creates oxidative damage to membranes and photosynthetic pigments [16]. In an attempt to defend against drought, plants may activate multiple morphophysiological, biochemical, and enzymatic responses, highlighting osmotic adjustment as the main mechanism for the large majority of plant species [17,18]. 

Numerous research centers have devoted intensive efforts to searching for methods to improve plant defense to water deficit. Efforts have been made by addressing plant breeding with different genes [19,20], and cross-talk tolerance inducers to increase the photosynthetic performance, antioxidant system, and promote osmotic adjustment [21,22,23]. Nonetheless, despite efforts to develop cultivars that combine high productivity with resistance to biotic and abiotic stresses, challenges remain in the search for cultivars with high ability under water limitation. These cultivars can decisively contribute to maintaining the agribusiness of large grain producers and the productive stability of small-scale farmers located in areas affected by drought episodes. Therefore, studies focusing on the selection of cultivars with elevated performance under low water availability become essential to selecting plants more resistant to drought occurrences.

Our working hypothesis was that soybean and cowpea cultivars display distinct responses to water deficit, which arise from biochemical adjustments to optimize plant performance. To test this hypothesis, seven semiarid-cultivated soybean and cowpea cultivars were exposed to different water availability levels under greenhouse conditions. Growth and some biochemical stress indicators were analyzed in both leguminous crops.

## 2. Results

### 2.1. Plant Growth

In general, soybean plants grown in soil with 60% field capacity (FC) exhibited values of shoot fresh mass (SFM) and shoot dry mass (SDM), root fresh mass (RFM), root dry mass (RDM), total fresh mass (TFM), and total dry mass (TDM) similar or higher than those of very well-irrigated plants (75% FC treatments) (Figure 1). All growth parameters were dramatically decreased by water limitation in soil, with rare exceptions (45 and 30% FC) (Figure 1 and Table 1). Also, in the majority of soybean cultivars, the fresh and dry mass decrease was intensified by reducing the irrigation water level (from 45 to 30% FC), and the reductions were found to be more prominent in the AS3810 IPRO, NS8338 IPRO BMX 81I81 IPRO, and M8808 IPRO cultivars (Figure 1 and Table 1). On the other hand, the lowest drought-induced reductions in plant growth were registered in BÔNUS8579 IPRO (at 45% FC) and TMG1180 RR (at 30% FC) cultivars, which exhibited the highest values of total dry mass and relative tolerance to drought as compared to other studied soybean cultivars (Figure 1 and Figure 2C,D). 

Water deprivation also promoted a strong decrease in the fresh and dry mass of tissues from all cowpea cultivars, with main reductions varying from 63 to 88% and 86 to 94% in plants growing under 60 and 45% FC as compared to the control (75% FC), respectively (Figure 3 and Table 2). At 30% FC, the drought deleterious effects were lethal for cowpea, and the plants exhibited severe symptoms of stress and died from the tenth to the fourteenth day of treatment. In absolute terms, drought-stressed Xique-xique plants displayed the highest values of shoot dry mass at 60% FC, whereas TVU plants showed the lowest ones at 45% FC, in relation to other cowpea cultivars (Figure 4B,C and Table 2). Similarly, the total dry mass under drought stress was higher in Xique-xique, Novaera and Pajeú (only at 60% FC) plants than in other cowpea cultivars at both levels of water limitation (60 and 45% FC). As a consequence of biomass accumulation, the Xique-xique, Novaera, and Pajeú plants exhibited the highest relative tolerance indexes for 60% FC treatments, respectively; whereas Xique-xique and Novaera were found to be the most tolerant under 45% FC (Figure 4B,C and Table 2). 

### 2.2. Relative Water Content and Membrane Damage

In soybean, relative water content (RWC) was significantly altered by water limitation only in the BÔNUS8579 IPRO cultivar, where plants growing under 45% FC displayed a 26.8% increase as compared to those from 75% FC treatment (Figure 1F,G and Table 1). Under drought (45% FC), the highest RWC values were registered in M8808 IPRO, NS8338 IPRO, and BÔNUS8579 IPRO in comparison to other studied cultivars (Figure 2C and Table 1). 

The membrane damage was significantly increased by water deficit in the leaf and roots of soybean plants as compared to respective controls (75% FC well-irrigated plants), depending on cultivar (Figure 1 and Table 1). In the leaves, the effects were more evident in the BMX 81I81 IPRO, M8808 IPRO, and BÔNUS8579 IPRO plants, while in roots they were observed in the TMG1180 RR, M8808 IPRO, and BÔNUS8579 IPRO ones. However, the drought-induced damages were more conspicuous in the roots from 30% FC treatments, where stressed M8808 IPRO and BÔNUS8579 IPRO plants showed values 389 and 327% higher than those of 75% FC-treated plants, respectively (Figure 1F,G and Figure 2D). 

Significant decreases in the RWC of cowpea were observed only in Pajeú, Tumucumaque, and TVU under 45% FC, and Xique-xique plants for both 60 and 45% FC treatments (Figure 3 and Table 2). Under drought, the highest RWC values were registered in Aracê and TVU plants grown at 60% FC, and in Novaera, Pitiúba, and Xique-xique cultivars at 45% FC (Figure 4B,C). Interestingly, the water restriction treatments did not increase the membrane damage to cowpea plants in comparison to well-irrigated plants (75% FC) (Figure 3), and significant alterations among drought-stressed cowpea cultivars were exclusively registered in leaves from plants grown under 45% FC treatments (Figure 4C and Table 2).

### 2.3. Accumulation of Photosynthetic Pigments

The photosynthetic pigments of soybean were found to be regulated differently in response to water deficit, depending on cultivar and stress level (Figure 1 and Table 1). Except for M8808, all soybean cultivars exhibited a significant decrease in content of chlorophyll (Chl) a under 45% and 30% drought treatments (Figure 1). Conversely, the cultivars AS3810 IPRO and M8644 IPRO under drought stress (45% and 30% FC) showed lower contents of Chl b and Chl total compared to their respective controls (75% FC) (Figure 1A,B and Table 1). Additionally, significant decreases in carotenoid levels were only observed in NS8338 IPRO, BMX 81I81 IPRO, and M8808 IPRO due to water deficit (Figure 1D–F and Table 1). Among the drought treatments, the cultivars M8644 IPRO and NS8338 IPRO exhibited the lowest contents of Chl a, Chl b, Chl total (only at 45% FC), and carotenoids (only at 45% FC) compared to other cultivars (Figure 2C,D and Table 1). 

In cowpea, drought stress (60% and 45% FC treatments) promoted a significant decrease in the Chl a content of cultivars Pitiúba, TVU, and Xique-xique. A similar response was only observed at 45% FC for the Aracê, Novaera, and Tumucumaque cultivars, compared to their respective controls (Figure 3 and Table 2). TVU plants showed a strong decrease in Chl b content due to drought treatment, with the response being intensified by the level of water restriction (Figure 3F). Novaera, Pitiúba, and Tumucumaque also exhibited decreases in Chl b content but at specific levels of water restriction (Figure 3A,D,E, and Table 2). Significant decreases in Chl total content were observed in Pitiúba and TVU plants exposed to 60% and 45% FC treatments, while it occurred in Aracê, Novaera, and Tumucumaque plants exposed to 45% FC treatments, compared to the control (Figure 3 and Table 2). In addition, carotenoid contents were found to be significantly decreased by all drought treatments in Pajeú, Pitiúba, TVU, and Xique-xique cultivars, whereas the decrease in Aracê, Novaera, and Tumucumaque was observed only at 45% FC, compared to their respective controls. Under moderate drought (60% FC), the highest contents of Chl a, Chl b, and total Chl were registered in the cultivars Tumucumaque, Novaera, and Aracê, compared to other cowpea cultivars (Figure 4 and Table 2). However, the Novaera cultivar was found to display the lowest contents of photosynthetic pigments under the 45% FC drought treatment, followed by Tumucumaque and TVU plants.

### 2.4. Accumulation of Organic Compounds

Soluble carbohydrates (SC) in soybean plants were found to be significantly reduced by drought in the M8644 IPRO and M8808 IPRO cultivars, at both 45% and 30% FC levels, when compared to the control (Figure 1B,F and Table 1). Under drought conditions at 45% FC, the highest SC contents were registered in AS3810 IPRO, M8644 IPRO, TMG 1180 RR, and BMX81I81 IPRO (Figure 2C and Table 1). On the other hand, at 30% FC, the highest values were observed in AS3810 IPRO and Bônus8579 IPRO, followed by M8644 IPRO, TMG 1180 RR, NS8338 IPRO, and BMX81I81 IPRO; M8808 IPRO exhibited the lowest SC accumulation (Figure 2D and Table 1).

For cowpea, drought stress caused a significant decrease in leaf SC content only in the Novaera and Tumucumaque cultivars, specifically at the 45% FC level, compared to the control (75% FC) (Figure 3B,E and Table 2). Furthermore, at 45% FC, the Pajeú, Pitiúba, and Xique-xique cultivars exhibited higher SC contents than the Aracê, Novaera, Tumucumaque, and TVU plants (Figure 4C and Table 2).

The free amino acids (AA) were differentially regulated by drought treatments and cultivars in both leguminous crops. In soybean, the TMG 1180 RR cultivar exhibited the most notable alterations, showing a significant increase in leaf AA under drought stress (Figure 1C and Table 1). It also displayed the highest AA content at 45% FC compared to other cultivars studied (Figure 2C and Table 1). Conversely, the BÔNUS8579 IPRO cultivar demonstrated decreased AA content at all drought levels studied, compared to their respective controls. In cowpea, the Aracê, Novaera, and Tumucumaque cultivars showed a significant increase in leaf AA content due to drought treatments, with the most prominent response observed in Aracê plants (Figure 3A,B,E and Appendix A). In contrast, Pajeú, Pitiúba, TVU, and Xique-xique plants exhibited a significant decrease in AA content under specific levels of drought stress, compared to the control (Figure 3). Additionally, at the 45% FC level, the highest AA accumulation was observed in Aracê plants, followed by Tumucumaque and Novaera, while the lowest values were found in the Pitiúba cultivar (Figure 4C and Table 2).

The leaf proline content of drought-stressed M8808 soybean plants was higher than that of well-irrigated plants, at both 45% and 30% FC levels (Figure 1F and Table 1). For other soybean cultivars, proline levels remained unchanged or decreased due to water restriction treatments. Under the 45% FC drought treatment, the highest proline levels were observed in the AS3810 IPRO, NS8338 IPRO, and M8808 plants (Figure 2C and Table 1); whereas the biggest proline accumulation at 30% FC level was registered in the NS8338 IPRO and M8808 cultivars (Figure 2D and Table 1).

The cowpea cultivars exhibited a significant increase in leaf proline content when subjected to 45% FC drought treatment, compared to well-irrigated plants at the 75% FC level, except Novaera and Pajeú plants (Figure 3 and Table 2). The accumulation of proline was more particularly evident in drought-stressed TVU plants, which displayed intensification of proline content by increasing water restriction (from 60 to 45% FC) (Figure 3F), resulting in the highest proline levels among the cowpea cultivars at 45% FC (Figure 4C and Table 2).

### 2.5. Principal Component Analysis (PCA)

Principal Component Analysis (PCA) was designed to investigate the correlation within soybean/cowpea crops and the parameters that best separated the cultivars for tolerance to drought (Figure 5). In soybean, the data explained 63.0% of total variation, with 48.3 and 14.7% explaining the first and second components, respectively (Figure 5A). The biplot analysis reveals an overlap between the 75 and 60% FC treatments, indicating similar performance, which distinguished them from the 45 and 30% FC treatments. At 45% FC, the cultivars BÔNUS8579 IPRO and TMG1180 RR exhibited the most remarkable responses, closer to well-irrigated plants. The growth parameters showed significant correlations with RWC, drought tolerance, soluble carbohydrates, and photosynthetic pigments, indicating a positive relationship with well-irrigated plants at 75% and 60% FC. Proline and amino acids were significantly correlated with some 45% FC-stressed plants, while root membrane damage showed a strong correlation with severe water restriction at 30% FC (Figure 5A).

For cowpea, the data explained 72.5% of total variation, it being 58.9 and 13.6% for the first and second components, respectively (Figure 5B). The cowpea biplot reveals a good separation between treatments based on water levels, suggesting a performance similar for cowpea plants under drought. Nevertheless, under water restriction, the most expressive responses were observed in drought-stressed Xique-xique plants under 60 and 45% FC which exhibited a closer performance to that of the control plants (Figure 5B). By investigating the Pearson correlation coefficients, growth parameters correlated significantly with photosynthetic pigments, RWC, and drought tolerance, demonstrating a positive correlation with 75% well-irrigated plants (Figure 5B). Soluble carbohydrate was closely related with 60% FC-stressed plants, whereas free amino acids correlated significantly with proline, displaying a high positive correlation with severely stressed plants (45% FC) (Figure 5B).

## 3. Discussion

### 3.1. Soybean Crop Has Low Water Requirements for Elevated Growth during the Vegetative Stage and Displays Drought Tolerance Higher Than Cowpea Crop

The data obtained from soybean crop revealed a similar performance between plants grown at 75% and 60% FC treatments. In total, 60% FC-grown plants showed values of fresh and dry biomass comparable (M8808 IPRO, AS3810 IPRO, BMX81I81 IPRO, and BÔNUS8579 IPRO) or higher (M8644 IPRO, TMG1180 RR, and NS8338 IPRO) than those of 75% FC-grown plants (Figure 1, Figure 2 and Figure 5). These results indicate that soybean plants did not experience water stress at 60% FC, and, in fact, this condition constitutes an ideal water availability for cultivation. Clearly, the studied soybean cultivars, cultivated in the semi-arid region, demonstrate a remarkable ability to withstand a significant level of water restriction. Furthermore, our findings suggest innovations for water-saving during the vegetative phase of the crop, a key claim for agricultural production considering future projections of water scarcity [24,25].

Both leguminous crops exhibited distinct patterns of response to water deficit, highlighting that soybean plants tolerate more severe levels of water restriction than cowpea plants. The evidence was that all cowpea cultivars displayed high sensitivity at 30% FC, exhibiting severe stress symptoms, and died between the tenth and fourteenth day of treatment. Therefore, analyzing the data, we defined that 60% FC represents well-irrigated conditions for soybean, 45% FC corresponds to a moderate drought, and 30% FC indicates a severe drought (Figure 1, Figure 2 and Figure 5A). In contrast, the data indicate that 75% FC is an ideal condition for cowpea cultivation, while the 60% and 45% FC treatments represent moderate and severe drought, respectively (Figure 3, Figure 4 and Figure 5B). These findings underscore that the ideal level of water availability depends on the crop and cultivar, and limited soil moisture can significantly impact plant growth and development.

### 3.2. Cowpea and Soybean Cultivars Display Contrasting Responses to Water Deficit

In the current study, under moderate (45%) and severe (30%) drought for soybean, the highest values of fresh and dry biomass were recorded in the BÔNUS8579 IPRO and TMG1180 RR plants, demonstrating the greatest relative tolerance to water deficit (Figure 2 and Figure 5A). On the contrary, under the same conditions, the M8808 IPRO, BMX81I81 IPRO, M8644 IPRO, and AS3810 IPRO cultivars exhibited the lowest biomass accumulation and were highly sensitive to water deficit (Figure 2 and Figure 5A).

Our data are consistent with previous studies in soybean cultivars, which report that water stress affects growth and carbon partition differently, since some cultivars are less tolerant compared to others [26,27]. Similarly, the highest drought tolerance was found in plants capable of maintaining phenotypic traits in a study investigating 20 soybean cultivars exposed to drought [28]. The authors emphasized the significant correlation between root characteristics and water stress tolerance, underscoring their crucial role in determining agronomic traits during vegetative growth.

The cowpea cultivars also displayed differential responses to the studied treatments, especially under moderate (60% FC) and severe drought (45% FC) (Figure 4 and Figure 5B). In both stress levels, the highest biomass values were recorded in the stressed Xique-xique and Novaera cultivars that demonstrated the highest levels of relative tolerance to drought (Figure 4B,C and Figure 5B). Conversely, the TVU plants showed the lowest biomass accumulation and stood out as the most drought-sensitive cultivar. These data are in concordance with previous studies which demonstrated that water deficit negatively impacts biomass accumulation, revealing a response that varies depending on the cultivar [29,30]. 

### 3.3. Leguminous Crops Activate Specific Biochemical Mechanisms for Drought Tolerance

Numerous reports have cited the loss of photosynthetic pigments as a primary signal of responses to water stress [14,24,31]. In agreement, our results revealed a significant decrease in Chl a, Chl b, Chl total, and carotenoids in almost all soybean and cowpea studied cultivars (Figure 1 and Figure 3 and Table 1 and Table 2). The decrease in chlorophyll levels may be a result of both the down regulation of biosynthetic pathways and increased hydrolysis by hydrolytic enzymes, as well as oxidative damage to the chloroplast membrane and chlorophyll degradation [31,32,33]. Herein, the accumulation of photosynthetic pigments in plants exposed to drought suggests the participation of differential mechanisms between leguminous crops. For soybean, a differential response among drought-contrasting stressed cultivars was practically absent, probably due to genetic potential to withstand water deficit impacts (Figure 2C,D and Table 1). On the other hand, for cowpea, a high content of photosynthetic pigments was recorded in the drought-tolerant Xique-xique cultivar and a low content was exhibited in the drought-sensitive TVU cultivar (Figure 4B,C and Figure 5B and Table 2). These data suggest a close relationship between chlorophyll accumulation and growth performance for cowpea plants. 

Our findings reinforce the crucial role of biochemical adjustments for soybean performance under drought treatments. The elevated sensitivity to drought observed in the M8808 soybean cultivar was associated with reduced accumulation of soluble carbohydrate, likely due to impaired CO_2_ assimilation under water limitation (Figure 1F) [33,34]. These implications suggest an energy excess at the PSII level and increased oxidative damage, as supported by the recorded membrane damage in leaves and roots (Figure 1F) [35,36,37]. In contrast, the TMG1180 RR cultivar, which exhibited greater tolerance to drought, showed a smaller decrease in biomass accumulation under drought stress, a response associated with the accumulation of free amino acids. These amino acids likely played a role in plant defense pathways such as osmotic adjustment (Figure 3 and Figure 5A) [38,39,40].

In cowpea, the good performance of the drought-tolerant Xique-xique cultivar was attributed to the maintenance of RWC and photosynthetic pigments, which acted to maintain the photosynthetic efficiency even at low water availability. This argument is corroborated by the unaltered level of soluble carbohydrates in the leaves of stressed plants (Figure 3G). Also, the proline accumulation in severely water-stressed Xique-xique plants might play an active role in the regulation of RWC and membrane damage for growth recovery, as previously reported for maize plants under salt stress (Figure 3G and Figure 5B) [41]. Yet, the Novaera cultivar activated a greater accumulation of free amino acids, probably to act in osmotic adjustment under water limitation (Figure 3B and Figure 5B). These results indicate a likely activation of control mechanisms to maintain water absorption and tissue water content, as well as to prevent harmful damage to cellular components.

Similarly, all stressed cowpea cultivars displayed an excessive accumulation of proline in the leaves, especially under severe drought, with a more pronounced response in drought-sensitive TVU plants (Figure 3 and Figure 5B). Proline is a versatile molecule that can elicit numerous defense responses, like osmotic adjustment, protein and membrane stabilization, scavenging of free radicals, signaling in cellular events, and gene expression [42]. Our results seem to indicate that proline serves as a molecular marker of drought stress, establishing a constitutive defense mechanism in cowpea plants against water deficit.

In general, our findings distinctly highlight pathways to counteract drought-related damages by identifying resilient cultivars and guiding targeted studies for developing effective agricultural advancements. These encompass the exploration of tolerance markers, crossbreeding methods, crosstalk tolerance inducers, and enhanced fertilizer management practices. Taken together, the progress can ensure both food security and environmental sustainability, particularly in regions prone to water scarcity. 

## 4. Materials and Methods

### 4.1. Plant Material and Growth Conditions

The experiments were carried out in a greenhouse at the Federal University of Piauí with geographic coordinates of 9°05′02.5″ S and 44°19′32.7″ W at approximately 650 m altitude. The analyses were performed at the Laboratories of the Campus Professora Cinobelina Elvas. The plants were grown in 11 dm³ plastic pots filled with soil from experimental site at UFPI (geographic coordinates of 9°04′45.6″ S and 44°19′37.9″ W), which was analyzed chemically and corrected [43]. During the trials, the environmental conditions inside the greenhouse were as follows: maximum and minimum temperatures of 27.4 ± 2.0 °C and 25.9 ± 2.2 °C, respectively; relative air humidity of approximately 68.5 ± 4.0%; and a photoperiod of approximately 12 h.

Two independent experiments were conducted in a randomized complete design, in 7 × 4 factorial schemes composed of seven soybean (*Glycine max* L.) or cowpea (*Vigna unguiculata* L. Walp) cultivars and four water level treatments [field capacity (FC) at 75, 60, 45, and 30%]. For soybean trials, the cultivars AS3810 IPRO, M8644 IPRO, TMG1180 RR, NS8338 IPRO, BMX81I81 IPRO, M8808 IPRO, and BÔNUS8579 IPRO were investigated; while the cultivars Aracê, Novaera, Pajeú, Pitiúba, Tumucumaque, TVU, and Xique-xique were studied in the cowpea trials. In all cases, four replications per treatment (one plant per pot) were employed, totaling 112 experimental units per experiment.

The soybean and cowpea cultivars were chosen based on their performance, production stability, and wide adaptability across the agricultural regions of the Brazilian Cerrado and semiarid areas. These regions are characterized by rainfed cultivation and subjected to periods of extended dry spells. The soybean seeds were acquired through Celeiro Sementes farm, a specialized company in soybean seed propagation. Cowpea seeds were obtained from the Active Germplasm Bank (https://av.cenargen.embrapa.br/avconsulta/Passaporte/detalhesBanco.do?idb=772, accessed on 8 March 2023) at the Federal University of Ceará (UFC), in Fortaleza, Ceará, Brazil. Detailed information about the soybean and cowpea cultivars are outlined in Appendix A, respectively. Before the sowing, the seeds were sterilized with 2% sodium hypochlorite and then sown in soils previously irrigated to 75% FC. At 14 days after sowing, uniform seedlings were selected and the drought treatments were imposed by reducing the water in irrigation to 60, 45, and 30% FC. A group of plants remained under irrigation of 75% FC, constituting the well-irrigated plants. The plants were watered daily to defined water levels using the weighing principle. The harvests were conducted 21 days after beginning the drought treatments, at the end of the vegetative stage.

### 4.2. Plant Growth

At harvest time, the plants were initially separated into leaves, stem, and roots to measure their fresh mass (FM). Subsequently, the plant material was immediately frozen and lyophilized to measure the dry mass. The index of relative tolerance to drought (RToler) was calculated by comparing the total dry mass of drought-stressed plants to the total dry mass of control plants [44]. Specifically, the total dry mass of plants grown under 60% FC for soybean and 75% FC for cowpea were used as controls for calculating the Rtoler index.

### 4.3. Relative Water Content and Membrane Damage

To estimate the relative water content (RWC), leaf discs were initially weighed to obtain the fresh weight (FW). Subsequently, the samples were immersed in deionized water for a period of 6 h at room temperature to achieve full turgidity, and the weight was recorded as the turgid weight (TW). Afterward, the samples were subjected to oven drying at 65 °C for 24 h and weighed to determine the dry weight (DW). The RWC was estimated by using the formula RWC (%) = [(FW – DW)/(TW – DW)] × 100.

Electrolyte leakage was employed as a measure of cell membrane damage, utilizing a conductivity meter. Leaf discs and root samples weighing 0.1 g were placed in sealed vials filled with deionized water, and then incubated at room temperature on a rotary shaker for 12 h. Thereafter, the electrical conductivity of the solution (EC1) was measured. Later, the homogenate was subjected to incubation at 100 °C for 15 min, and the conductivity was measured once more (EC2). The percentage of membrane damage (%) was calculated as EC1/C2 × 100.

### 4.4. Photosynthetic Pigments

The photosynthetic pigments were extracted by incubating leaf discs in dimethyl sulfoxide (DMSO) saturated with CaCO_3_ at room temperature in the dark for 72 h. The contents of chlorophyll a (Chl a), chlorophyll a (Chl b), chlorophyll total (Chl total), and carotenoids were estimated through spectrophotometry readings at wavelengths of 480, 649, and 665 nm, following the equations defined by Wellburn method [45].

### 4.5. Soluble Carbohydrates, Free Amino Acids and Proline

Soluble carbohydrates and free amino acids were extracted after homogenizing 10 mg of leaf lyophilized samples in 5 mL of 80% aqueous ethanol at 75 °C for 1.0 h. Then, the homogenate was centrifuged at 3000× *g* for 10 min at 4 °C, and the resulting supernatant was collected. This extraction process was repeated twice on the remaining precipitate. Subsequently, all the collected supernatants were combined and adjusted to a final volume of 25 mL with 80% ethanol. The content of soluble carbohydrates was estimated by readings at 490 nm, using anhydrous D-glucose as a standard [46]. Yet, the content of free amino acids was measured by spectrophotometric readings at 570 nm, using L-glycine as a standard [47]. 

The proline content was assessed using the ninhydrin method as outlined by the Bates method [48]. Aqueous extracts were prepared by homogenizing 20 mg of lyophilized leaves in 2.0 mL of deionized water at 75 °C for 1.0 h. Thereafter, the homogenate was centrifuged at 3000× *g* for 10 min at room temperature. The supernatant was collected and used to measure the proline content by absorbance at 520 nm, with L-proline serving as the standard.

### 4.6. Statistical Analysis

For each experiment, the data underwent analysis of variance (ANOVA) using the F test at a significance level of 5%. Post hoc comparisons of means were conducted using Scott–Knott’s test (*p* ≤ 0.05) with the Sisvar software [49]. The clustering analyzes were designed using Excel software, and the radar plot graphs were plotted through Sigma Plot 11.0 software (SPSS Inc., San Jose, California, USA). Additionally, principal component analysis (PCA) was performed on the datasets by using the R software [50].

## 5. Conclusions

Our findings reveal insights into innovative agricultural practices, underscoring the remarkable efficiency of soybean in managing water requirements, not only in terms of water-saving potential in agricultural production but also in terms of enhanced production strategies. TMG1180 RR and BÔNUS8579 IPRO are the most drought-resistant soybean cultivars, highlighting defense pathways to avoid tissue desiccation. Xique-xique emerged as the most drought-resistant cowpea cultivar, characterized by a high ability to maintain water content, proline, and photosynthetic pigments, which contributed to the preservation of soluble carbohydrates under water restriction. The data might help plant breeders and farmers in mitigating drought-related damages in general, providing suitable information regarding tolerant soybean/cowpea cultivars for advancing plant breeding efforts, and exploring viable alternatives for cultivating leguminous crops in arid and semiarid regions toward sustainable and resilient agriculture.

## Figures and Tables

**Figure 1 plants-12-03134-f001:**
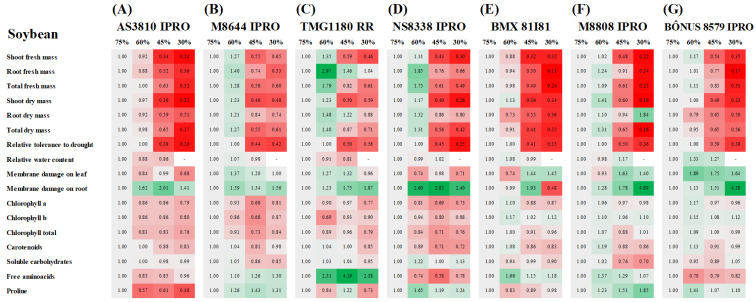
Clustering analysis of growth and biochemical assays relative to changes due to drought treatments in soybean cultivars: AS3810 IPRO (**A**), M8644 IPRO (**B**), TMG1180 RR (**C**), NS8338 IPRO (**D**), BMX81I81 IPRO (**E**), M8808 IPRO (**F**), and BÔNUS8579 IPRO (**G**). The trials were carried out in plants 21 days after exposure to four water level treatments (75, 60, 45, and 30% field capacity—FC). Each row characterizes an individual analysis. For all cases, green color specifies an increase, and red denotes a decrease in the analyzed indexes, taking the data of 75% FC plants as reference. Gray represents no change. The number inside the box and different red and green intensities express the extent of the change according to fold increase or decrease related to reference. For relative tolerance to drought, the total dry mass of plants from the 60% treatment was used as the control, and the plants from 45% to 30% FC were considered water deficit. Sufficient material was not obtained to calculate RWC at 30% FC. For absolute values and statistical details, see Table 1.

**Figure 2 plants-12-03134-f002:**
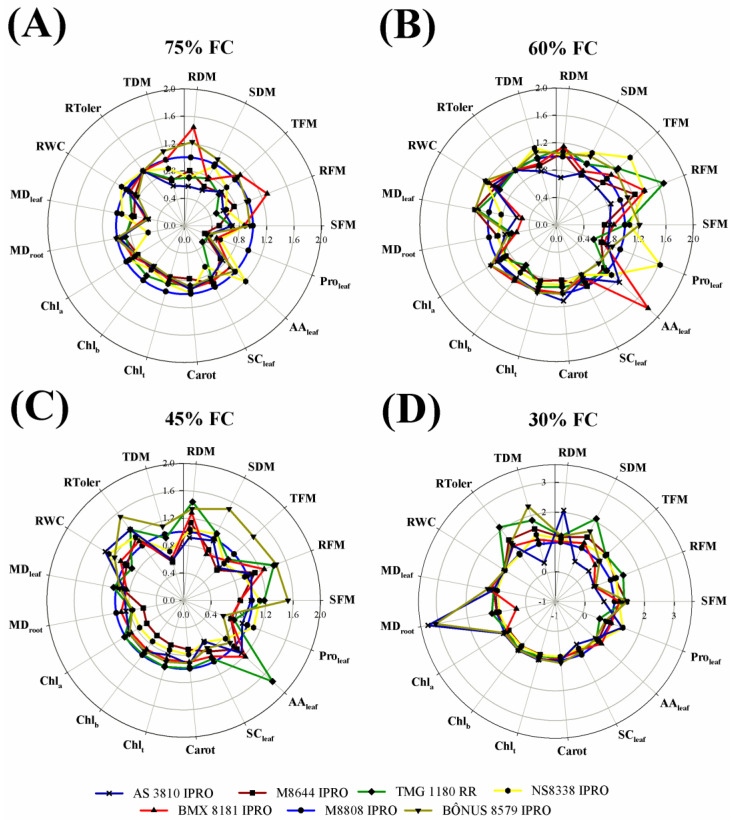
Overall representation of contrasting responses from *Glycine max* L. cultivars, AS3810 IPRO (dark blue line), M8644 IPRO (dark red line), TMG1180 RR (green line), NS8338 IPRO (yellow line), BMX81I81 IPRO (red line), M8808 IPRO (blue line), and BÔNUS8579 IPRO (dark yellow line), subject to different water levels treatments: 75 (**A**), 60 (**B**), 45 (**C**), and 30% (**D**) field capacity (FC). The data refer to relative alterations in the following parameters: shoot (SFM), root (RFM), and total fresh mass (TFM); shoot (SDM), root (RDM), and total dry mass (TDM); relative tolerance to drought (RToler); relative water content (RWC); membrane damage in leaf (MD_leaf_) and roots (MD_root_); contents of chlorophyll a (Chl_a_), b (Chl_b_), total (Chl_t_), and carotenoids (Carot); contents of soluble carbohydrates (SC_leaf_), free amino acids (AA_leaf_), and proline (Pro_leaf_) in the leaves. The radar plot was designed using the data of the M8808 cultivar (blue line) as a reference.

**Figure 3 plants-12-03134-f003:**
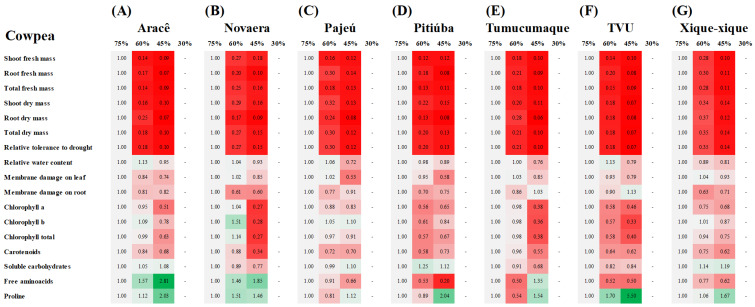
Clustering analysis of growth and biochemical assays relative to changes due to drought treatments in cowpea cultivars: Aracê (**A**), Novaera (**B**), Pajeú (**C**), Pitiúba (**D**), Tumucumaque (**E**), TVU (**F**), and Xique-xique (**G**). The trials were carried out in plants 21 days after exposure to four water levels treatments (75, 60, 45, and 30% field capacity—FC). Each row characterizes an individual analysis. For all cases, green color specifies an increase, and red denotes a decrease in the analyzed indexes, taking the data of 75% FC plants as reference. Cowpea plants did not support the 30% FC drought level and died before the harvest. Gray represents no change. Numbers inside the box and different red and green intensities express the extent of the change according to fold increase/decrease related to reference. For absolute values and statistical details, see Table 2.

**Figure 4 plants-12-03134-f004:**
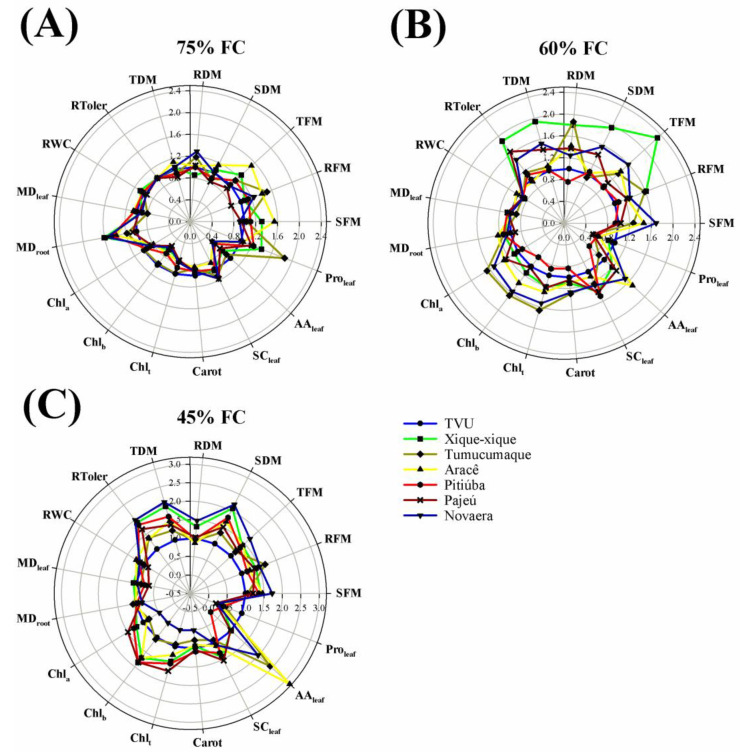
Overall representation of contrasting responses from *Vigna unguiculata* L. Walp cultivars, Aracê (yellow line), Novaera (dark blue line), Pajeú (dark red line), Pitiúba (red line), Tumucumaque (dark yellow line), TVU (blue line), and Xique-xique (green line), subject to different water levels treatments: 75 (**A**), 60 (**B**), and 45% (**C**) field capacity (FC). The data refer to relative alterations in the following parameters: shoot (SFM), root (RFM), and total fresh mass (TFM); shoot (SDM), root (RDM), and total dry mass (TDM); relative tolerance to drought (RToler); relative water content (RWC); membrane damage in leaf (MD_leaf_) and roots (MD_root_); contents of chlorophyll a (Chl_a_), b (Chl_b_), total (Chl_t_), and carotenoids (Carot); contents of soluble carbohydrates (SC_leaf_), free amino acids (AA_leaf_), and proline (Pro_leaf_) in the leaves. The radar plot was designed using the data of TVU cultivar (blue line) as a reference.

**Figure 5 plants-12-03134-f005:**
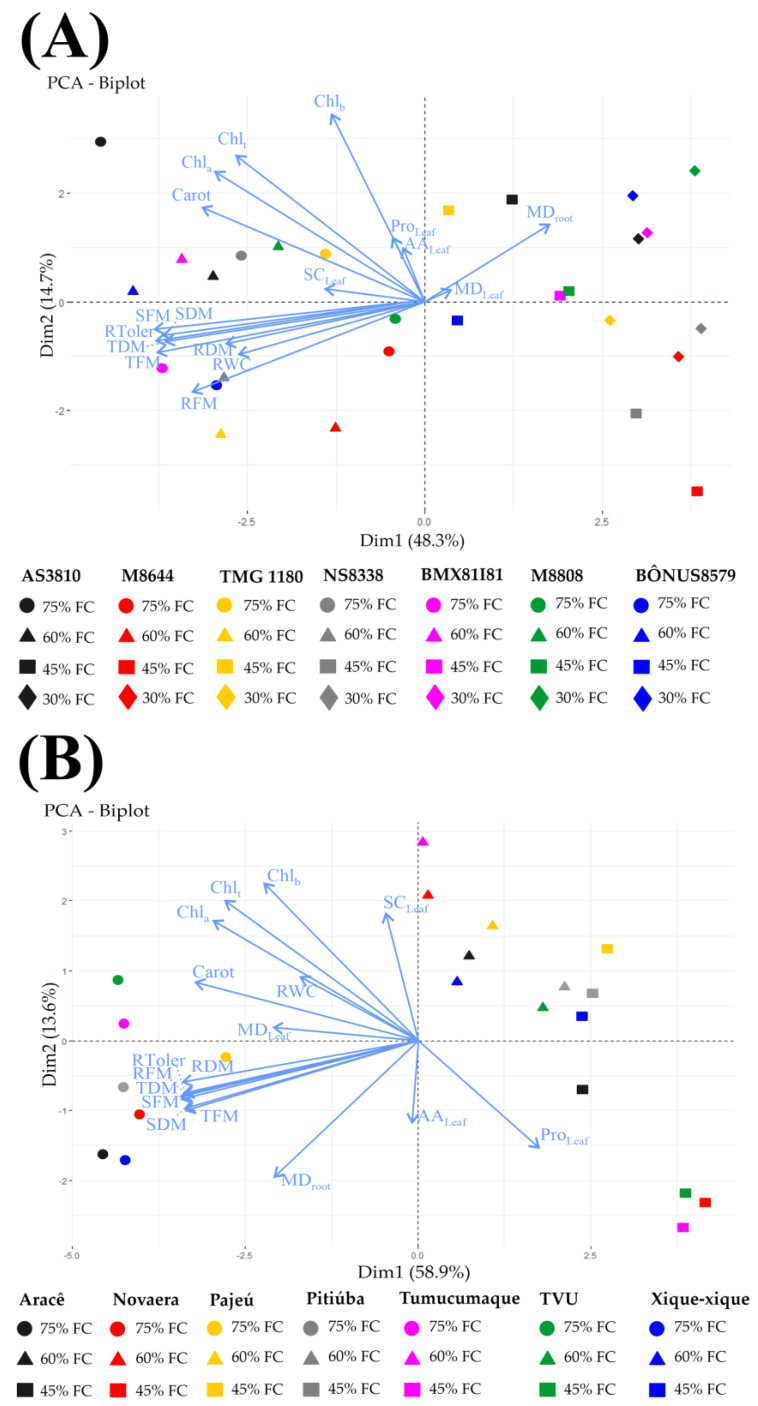
Principal Component Analysis (PCA). Scatter plots of parameters investigated in soybean (**A**) and cowpea (**B**) cultivars under different water availability levels (75, 60, 45, and 30% of field capacity—FC). The X and Y axes indicate the percentage of variance explained by each Principal Component (PC). The loading plot displays the contribution of the following parameters: shoot fresh mass (SFM), root fresh mass (RFM), and total fresh mass (TFM); shoot dry mass (SDM), root dry mass (RDM), and total dry mass (TDM); relative tolerance to drought (RToler); relative water content (RWC); membrane damage in leaves (MD_leaf_) and roots (MD_root_); contents of chlorophyll a (Chl_a_), chlorophyll b (Chl_b_), total chlorophyll (Chl_t_), and carotenoids (Carot); contents of soluble carbohydrates (SC_leaf_), free amino acids (AA_leaf_), and proline (Pro_leaf_) in the leaves.

**Table 1 plants-12-03134-t001:** Absolute values and statistical details for fresh and dry mass, relative tolerance to drought, membrane damage, photosynthetic pigments, soluble carbohydrates, free amino acids, and proline of soybean plants. The assays were carried out on cultivars of AS3810 IPRO, M8644 IPRO, TMG1180 RR, NS8338 IPRO, BMX 81I81 IPRO, M8808 IPRO, and BÔNUS8579 IPRO, 21 days after exposure to four water levels treatments (75, 60, 45, and 30% field capacity—FC).

Soybean Cultivars	75%	60%	45%	30%	75%	60%	45%	30%
	**Shoot fresh mass** (g/plant)				**Shoot dry mass** (g/plant)			
AS3810 IPRO	8.26 Aa	7.57 Aa	2.80 Ba	1.94 Ba	1.71 Aa	1.66 Aa	0.61 Ba	0.37 Ba
M8644 IPRO	4.20 Ac	5.32 Ab	2.33 Ba	2.72 Ba	1.09 Ab	1.35 Ab	0.50 Ba	0.53 Ba
TMG1180 RR	5.61 Bb	7.60 Aa	3.30 Ca	2.57 Ca	1.33 Ab	1.64 Aa	0.67 Ba	0.79 Ba
NS8338 IPRO	7.29 Aa	8.10 Aa	3.11 Ba	2.20 Ba	1.65 Aa	1.94 Aa	0.66 Ba	0.43 Ba
BMX81I81 IPRO	7.20 Aa	6.36 Ab	2.31 Ba	2.27 Ba	1.29 Ab	1.45 Ab	0.46 Ba	0.44 Ba
M8808 IPRO	5.77 Ab	5.89 Ab	2.77 Ba	1.24 Ca	0.99 Bb	1.40 Ab	0.60 Ca	0.18 Da
BÔNUS8579 IPRO	7.89 Aa	9.20 Aa	4.27 Ba	2.76 Ca	1.85 Aa	1.85 Aa	0.91 Ba	0.61 Ba
	**Root fresh mass** (g/plant)				**Root dry mass** (g/plant)			
AS3810 IPRO	4.07 Ab	3.58 Ab	2.11 Ba	1.45 Ba	0.44 Ac	0.40 Aa	0.26 Bb	0.23 Bb
M8644 IPRO	3.47 Bb	4.94 Aa	2.34 Ca	1.67 Ca	0.39 Ac	0.47 Aa	0.30 Ba	0.23 Bb
TMG1180 RR	2.03 Cc	6.03 Aa	2.96 Ba	2.10 Ca	0.34 Bd	0.46 Aa	0.37 Bb	0.27 Cb
NS8338 IPRO	2.94 Bc	4.92 Aa	2.01 Ca	1.74 Ca	0.31 Bd	0.41 Aa	0.27 Bb	0.25 Bb
BMX81I81 IPRO	5.27 Aa	4.95Aa	2.65 Ba	0.58 Cb	0.70 Aa	0.46 Ba	0.33 Ca	0.23 Db
M8808 IPRO	2.47 Ac	3.05 Ab	2.23 Aa	0.59 Bb	0.25 Bd	0.28 Bb	0.24 Bb	0.42 Aa
BÔNUS8579 IPRO	3.97 Ab	3.99 Ab	3.05 Ba	0.67 Cb	0.54 Ab	0.47 Aa	0.35 Ba	0.28 Bb
	**Total fresh mass** (g/plant)				**Total dry mass** (g/plant)			
AS3810 IPRO	12.33 Aa	11.15 Ac	4.90 Bc	3.40 Ba	2.15 Aa	2.27 Aa	0.87 Ba	0.60 Bb
M8644 IPRO	7.68 Bc	10.26 Ac	4.67 Cc	4.39 Ca	1.49 Ab	1.81 Ab	0.80 Ba	0.76 Bb
TMG1180 RR	7.64 Bc	13.62 Aa	6.26 Bb	4.27 Ca	1.67 Bb	2.10 Aa	1.04 Ca	1.17 Ca
NS8338 IPRO	10.23 Bb	13.02 Ab	5.12 Cc	3.94 Ca	1.96 Ba	2.35 Aa	0.93 Ca	0.60 Cb
BMX81I81 IPRO	13.59 Aa	12.14 Ab	4.96 Bc	2.85 Cb	2.28 Aa	1.91 Bb	0.79 Ca	0.67 Cb
M8808 IPRO	7.23 Bc	8.95 Ad	5.00 Cc	1.83 Db	1.24 Bb	1.68 Ab	0.83 Ca	0.60 Cb
BÔNUS8579 IPRO	11.86 Ba	14.47 Aa	7.94 Ca	3.43 Da	2.39 Aa	2.32 Aa	1.26 Ba	0.89 Ca
	**Relative tolerance to drought** (%) *				**Relative water content** (%) **			
AS3810 IPRO	100 Aa	100 Aa	38.39 Bc	26.34 Cc	52.6 Aa	46.3 Aa	45.4 Ab	-
M8644 IPRO	100 Aa	100 Aa	44.06 Bc	41.76 Bb	44.9 Aa	48.2 Aa	44.2 Ab	-
TMG1180 RR	100 Aa	100 Aa	49.64 Bb	55.86 Ca	49.5 Aa	45.2 Aa	40.3 Ab	-
NS8338 IPRO	100 Aa	100 Aa	44.64 Bc	25.43 Cc	56.9 Aa	56.1 Aa	58.3 Aa	-
BMX81I81 IPRO	100 Aa	100 Aa	41.41 Bc	35.07 Cb	49.3 Aa	53.5 Aa	49.0 Ab	-
M8808 IPRO	100 Aa	100 Aa	49.65 Bb	35.80 Cb	52.0 Aa	50.7 Aa	61.1 Aa	-
BÔNUS8579 IPRO	100 Aa	100 Aa	58.75 Ba	38.54 Cb	42.5 Ba	56.6 Aa	53.9 Aa	-
	**Membrane damage in leaves** (%)				**Membrane damage in roots** (%)			
AS3810 IPRO	30.4 Aa	25.7 Aa	30.1 Ba	20.6 Ba	35.9 Ba	58.1 Aa	72.1 Aa	50.6 Bb
M8644 IPRO	22.8 Bb	31.1 Aa	27.4 Aa	22.8 Ba	32.2 Aa	51.1 Aa	43.2 Aa	50.3 Ab
TMG1180 RR	23.7 Bb	30.1 Aa	31.3Aa	22.8 Ba	31.5 Ba	38.1 Ba	55.2 Aa	58.9 Ab
NS8338 IPRO	28.4 Aa	21.1 Bb	27.7 Aa	20.3 Ba	19.4 Ba	50.5 Aa	55.0 Aa	46.6 Ab
BMX81I81 IPRO	17.6 Bc	13.1 Bc	25.4 Aa	25.5 Aa	34.7 Ba	34.3 Ba	66.9Aa	16.5 Bc
M8808 IPRO	16.3 Bc	15.2 Bc	26.6 Aa	22.9 Aa	34.6 Ca	44.3 Ca	61.8 Ba	169.3 Aa
BÔNUS8579 IPRO	16.4 Bc	31.1 Aa	28.7 Aa	26.9 Aa	36.5 Ba	41.1 Ba	54.9 Ba	156.2 Aa
	**Chl a** (μg g^−1^ DM)				**Chl b** (μg g^−1^ DM)			
AS3810 IPRO	4419 Aa	3801 Bb	3821 Ba	3478 Ba	1376 Aa	1182 Ba	1180 Ba	1095 Ba
M8644 IPRO	3556 Ab	3289 Ac	2425 Bb	2896 Bb	1108 Ab	952 Bb	758 Bb	959 Bb
TMG1180 RR	3990 Aa	3603 Ac	3875 Aa	3059 Bb	1263 Aa	873 Bb	1169 Aa	1132 Aa
NS8338 IPRO	4179 Aa	3399 Bc	2883 Bb	3043 Bb	1106 Ab	1035 Ab	886 Ab	975 Ab
BMX81I81 IPRO	3888 Ab	4289 Aa	3436 Ba	3368 Ba	1036 Ab	1215 Aa	1057 Aa	1156 Aa
M8808 IPRO	3658 Ab	3881 Ab	3557 Aa	3596 Aa	1045 Ab	1145 Aa	1112 Aa	1153 Aa
BÔNUS8579 IPRO	3657 Bb	4279 Aa	3565 Ba	3499 Ba	1013 Ab	1163 Aa	1097 Aa	1134 Aa
	**Chl total** (μg g^−1^ DM)				**Carotenoids** (μg g^−1^ DM)			
AS3810 IPRO	6039 Aa	5016 Ba	5035 Ba	4602 Ba	951 Aa	950 Aa	837 Aa	787 Aa
M8644 IPRO	4695 Ab	4278 Aa	3408 Bb	3934 Ba	740 Aa	772 Ab	603 Ab	725 Aa
TMG1180 RR	5286 Aa	4722 Aa	5082 Aa	4200 Aa	836 Aa	870 Ab	835 Aa	712 Aa
NS8338 IPRO	5343 Aa	4466 Ba	3795 Bb	4042 Ba	942 Aa	836 Ab	665 Bb	676 Ba
BMX81I81 IPRO	4977 Ab	4960 Aa	4525 Aa	4780 Aa	882 Aa	955 Aa	759 Ba	735 Ba
M8808 IPRO	4744 Ab	5067 Aa	4177 Aa	4778 Aa	893 Ba	1060 Aa	784 Ba	772 Ba
BÔNUS8579 IPRO	4716 Ab	5154 Aa	4695 Aa	4660 Aa	853 Ba	962 Aa	774 Aa	845 Aa
	**Soluble carbohydrates** (μmol g^−1^ DM)				**Free amino acids** (μmol g^−1^ DM)			
AS3810 IPRO	1287 Aa	1283 Aa	1266 Aa	1275 Aa	217.8 Ab	180.1 Ab	184.1 Ac	210.1 Aa
M8644 IPRO	1228 Aa	1287 Aa	1058 Ba	1048 Bb	154.2 Ac	169.0 Ab	194.5 Ab	200.5 Aa
TMG1180 RR	1163 Aa	1199 Aa	1206 Aa	1100 Ab	77.4 Cb	178.6 Bb	324.1 Aa	183.9 Ba
NS8338 IPRO	863 Ab	1050 Ab	863 Ab	973 Ab	263.2 Aa	195.9 Bb	153.0 Cc	205.5 Ba
BMX81I81 IPRO	1173 Aa	1101 Ab	1157 Aa	1057 Ab	196.6 Bb	326.5 Aa	225.5 Bb	232.1 Ba
M8808 IPRO	1106 Aa	1124 Ab	845 Bb	774 Bc	163.9 Bc	224.4 Ab	211.2 Ab	175.9 Ba
BÔNUS8579 IPRO	1101 Aa	1042 Ab	984 Ab	1152 Aa	214.8 Ab	150.5 Bb	169.4 Bc	177.1 Ba
	**Proline** (μmol g^−1^ DM)							
AS3810 IPRO	6.69 Aa	3.79 Bb	4.07 Ba	3.24 Bb				
M8644 IPRO	2.11 Ac	2.67 Ab	3.02 Ab	2.76 Ab				
TMG1180RR	2.65 Ac	2.22 Ab	3.23 Ab	1.95 Ab				
NS8338 IPRO	3.75 Bb	6.17 Aa	4.46 Ba	4.66 Ba				
BMX81I81 IPRO	3.37 Ab	2.82 Ab	3.00 Ab	3.30 Ab				
M8808 IPRO	2.48 Bc	3.05 Bb	3.74 Aa	4.58 Aa				
BÔNUS8579 IPRO	2.36 Ac	3.33 Ab	2.53 Ab	2.61 Ab				

In the same line, different capital letters represent significant differences due to drought stress within the same soybean cultivar. In the same column, different lowercase letters represent significant alterations among soybean cultivars within the same stress level, according to Scott–Knott’s test (*p* < 0.05). * For relative tolerance to drought, the total dry mass of plants from the 60% treatment was used as the control, and the plants 45% and 30% FC were considered water deficit. ** Sufficient material was not obtained to calculate RWC at 30% FC.

**Table 2 plants-12-03134-t002:** Absolute values and statistical details for fresh and dry mass, relative tolerance to drought, membrane damage, photosynthetic pigments, soluble carbohydrates, free amino acids, and proline of cowpea plants. The assays were carried out on cultivars of Aracê, Novaera, Pajeú, Pitiúba, Tumucumaque, TVU, and Xique-xique, 21 days after exposure to four water level treatments (75, 60, 45, and 30% field capacity—FC). Cowpea plants did not support the 30% FC drought level and died before the harvest.

Cowpea Cultivars	75%	60%	45%	30%	75%	60%	45%	30%
	**Shoot fresh mass** (g/plant)				**Shoot dry mass** (g/plant)			
Aracê	64.83 Aa	8.83 Ba	5.96 Ba	-	7.18 Aa	1.18 Ba	0.75 Ba	-
Novaera	38.20 Ab	10.20 Ba	7.02 Ba	-	6.19 Aa	1.79 Ba	0.98 Ba	-
Pajeú	39.32 Ab	6.29 Ba	4.90 Ba	-	5.09 Aa	1.61 Ba	0.68 Ba	-
Pitiúba	46.48 Ab	5.69 Ba	5.55 Ba	-	5.54 Aa	1.20 Ba	0.81 Ba	-
Tumucumaque	43.41 Ab	7.683 Ba	4.30 Ba	-	5.42 Aa	1.07 Ba	0.60 Ba	-
TVU	42.02 Ab	6.04 Ba	4.07 Ba	-	6.21 Aa	1.15 Ba	0.45 Ba	-
Xique-xique	55.22 Aa	15.37 Ba	5.76 Ba	-	6.56 Aa	2.24 Ba	0.93 Ba	-
	**Root fresh mass** (g/plant)				**Root dry mass** (g/plant)			
Aracê	13.91 Aa	2.41 Ba	1.02 Ba	-	1.14 Ab	0.29 Ba	0.08 Ba	-
Novaera	12.35 Aa	2.51 Ba	1.25 Ba	-	1.45 Aa	0.25 Ba	0.13 Ba	-
Pajeú	7.93 Ab	2.40 Ba	1.13 Ba	-	1.16 Ab	0.28 Ba	0.09 Ba	-
Pitiúba	11.26 Ab	2.08 Ba	0.95 Ba	-	1.13 Ab	0.15 Ba	0.09 Ba	-
Tumucumaque	14.88 Aa	3.08 Ba	1.38 Ba	-	1.33 Aa	0.37 Ba	0.08 Ca	-
TVU	9.86 Ab	1.94 Ba	0.82 Ba	-	1.12 Ab	0.20 Ba	0.09 Ba	-
Xique-xique	10.55 Ab	3.12 Ba	1.16 Ba	-	0.96 Ab	0.36 Ba	0.12 Ba	-
	**Total fresh mass** (g/plant)				**Total dry mass** (g/plant)			
Aracê	78.75 Aa	11.23 Ba	6.97 Ba	-	8.33 Aa	1.46 Bb	0.82 Bb	-
Novaera	50.55 Ac	12.70 Ba	8.27 Ba	-	7.65 Aa	2.04 Ba	1.11 Ba	-
Pajeú	47.25 Ac	8.69 Ba	6.03 Ba	-	6.25 Aa	1.88 Ba	0.77 Bb	-
Pitiúba	57.74 Ac	7.766 Ba	6.50 Ba	-	6.67 Aa	1.36 Bb	0.89 Bb	-
Tumucumaque	58.29 Ac	10.76 Ba	5.69 Ba	-	6.75 Aa	1.44 Bb	0.68 Bb	-
TVU	51.88 Ac	7.97 Ba	4.89 Ba	-	7.33 Aa	1.35 Bb	0.54 Bb	-
Xique-xique	65.77 Ab	18.49 Ba	6.93 Ca	-	7.53 Aa	2.60 Ba	1.05 Ba	-
	**Relative tolerance to drought** (%) *				**Relative water content** (%)			
Aracê	100 Aa	10.1 Bb	8.3 Bb	-	84.5% Ba	95.2% Aa	80.2% Ba	-
Novaera	100 Aa	28.7 Ba	15.8 Ba	-	78.5% Aa	81.5% Ab	72.7% Aa	-
Pajeú	100 Aa	32.4 Ba	13.8 Cb	-	77.2% Aa	81.7% Ab	55.7% Bb	-
Pitiúba	100 Aa	13.7 Bb	21.2 Ba	-	84.2% Aa	82.5% Ab	75.5% Aa	-
Tumucumaque	100 Aa	12.8 Bb	9.0 Bb	-	83.0% Aa	82.7% Ab	63.0% Bb	-
TVU	100 Aa	18.9 Bb	7.6 Bb	-	82.5% Aa	92.7% Aa	65.2% Bb	-
Xique-xique	100 Aa	36.7 Ba	15.5 Ca	-	89.2% Aa	79.2% Bb	72.2% Ba	-
	**Membrane damage in leaves** (%)				**Membrane damage in roots** (%)			
Aracê	82.5% Aa	69.0% Ba	61.5% Ba	-	60.5% Aa	48.7% Ba	49.5% Ba	-
Novaera	70.0% Ab	72.0% Aa	59.5% Ba	-	68.0% Aa	42.0% Ba	41.2% Ba	-
Pajeú	74.2% Aa	76.2% Aa	39.7% Bb	-	47.0% Ab	36.2% Aa	43.0% Aa	-
Pitiúba	80.5% Aa	76.2% Aa	47.0% Bb	-	64.5% Aa	45.0% Ba	48.0% Ba	-
Tumucumaque	62.0% Ab	64.2% Aa	53.0% Ab	-	51.7% Ab	44.5% Aa	53.5% Aa	-
TVU	77.7% Aa	72.2% Aa	61.7% Ba	-	44.0% Ab	39.7% Aa	49.2% Aa	-
Xique-xique	70.7% Ab	73.7% Aa	66.0% Aa	-	70.5% Aa	44.2% Ba	50.0% Ba	-
	**Chl a** (μg g^−1^ DM)				**Chl b** (μg g^−1^ DM)			
Aracê	3353 Ab	2535 Ab	1366 Bb	-	1084 Ab	1181 Aa	847 Aa	-
Novaera	2858 Ab	2959 Aa	764 Bc	-	892 Bb	1351 Aa	252 Cb	-
Pajeú	2785 Ab	2461 Ab	2324 Aa	-	841 Ab	883 Ab	921 Aa	-
Pitiúba	3144 Aa	1769 Bc	2030 Ba	-	1101 Ab	670 Bb	925 Aa	-
Tumucumaque	3335 Aa	3256 Aa	1273 Bb	-	1459 Aa	1427 Aa	532 Bb	-
TVU	3353 Aa	1958 Bc	1559 Bb	-	1504 Aa	858 Bb	501 Cb	-
Xique-xique	2777 Ab	2095 Bc	1893 Ba	-	974 Ab	984 Ab	849 Aa	-
	**Chl total** (μg g^−1^ DM)				**Carotenoids** (μg g^−1^ DM)			
Aracê	3744 Ab	3699 Ab	2375 Ba	-	492 Aa	414 Ab	332 Ba	-
Novaera	3775 Ab	4292 Aa	1022 Bc	-	542 Aa	479 Aa	185 Bc	-
Pajeú	3573 Ab	3464 Ab	3245 Aa	-	543 Aa	393 Bb	379 Ba	-
Pitiúba	4261 Aa	2442 Bc	2845 Ba	-	541 Aa	313 Bb	396 Ba	-
Tumucumaque	4780 Aa	4669 Aa	1803 Bb	-	510 Aa	489 Aa	282 Bb	-
TVU	4839 Aa	2808 Bc	1922 Bb	-	587 Aa	375 Bb	365 Ba	-
Xique-xique	3632 Ab	3404 Ab	2732 Aa	-	552 Aa	415 Bb	344 Ba	-
	**Soluble carbohydrates** (μmol g^−1^ DM)				**Free amino acids** (μmol g^−1^ DM)			
Aracê	194 Ab	204 Ab	209 Ab	-	535 Cb	843 Ba	1504 Aa	-
Novaera	267 Aa	238 Aa	204 Bb	-	514 Bb	753 Aa	949 Ac	-
Pajeú	268 Aa	264 Aa	295 Aa	-	703 Ab	592 Aa	466 Bd	-
Pitiúba	224 Ab	280 Aa	250 Aa	-	583 Ab	310 Bb	119 Be	-
Tumucumaque	256 Aa	233 Aa	175 Bb	-	861 Ba	429 Cb	1160 Ab	-
TVU	229 Ab	187 Ab	192 Ab	-	959 Aa	496 Bb	477 Bd	-
Xique-xique	229 Ab	263 Aa	273 Aa	-	767 Aa	643 Ba	478 Bd	-
	**Proline** (μmol g^−1^ DM)							
Aracê	65 Ba	72 Ba	131 Ab	-				
Novaera	64 Aa	97 Aa	93 Ac	-				
Pajeú	75 Aa	61 Aa	84 Ac	-				
Pitiúba	77 Ba	69 Ba	158 Ab	-				
Tumucumaque	115 Ba	62 Ca	177 Ab	-				
TVU	62 Ca	105 Ba	341 Aa	-				
Xique-xique	87 Ba	92 Ba	145 Ab	-				

In the same line, different capital letters represent significant differences due to drought stress within the same soybean cultivar. In the same column, different lowercase letters represent significant alterations among soybean cultivars within the same stress level, according to Scott–Knott’s test (*p* < 0.05). * For relative tolerance to drought, the total dry mass of plants from the 75% treatment was used as the control, and the plants 60% and 45% FC were considered water deficit.

## Data Availability

Not applicable.

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
