# Peer review of "Selection of Soybean and Cowpea Cultivars with Superior Performance under Drought Using Growth and Biochemical Aspects"

_plants, 2023, doi:10.3390/plants12173134_

Round 1

Reviewer 1 Report

The manuscript entitled "Selection of soybean and cowpea genotypes with superior performance under drought using growth and biochemical aspects" was well written. The objective was clear and the experiments were well-designed and the data were well-analyzed. Minor revision is needed before acceptance. 

1. The abbreviations should be explained in both abstract and main texts.

2. Page numbers are not correct. 

3. Line 228 is not clear.

4. The order of the figures was suggested to be reorganized based on the results in the texts. For example, in Section 2.1, the figures referred to in the texts followed the order 1, 3, 2, 4.  Similarly, in Section 2.2, the Figures referred to followed the order 1, 3, 4, 2... It is hard to find the figure accordingly. 

Author Response

We thank the excellent comments of the Reviewers, which we believe that they greatly contributed to improve the quality of manuscript. We have completed all of the changes requested and a long list with changes and responses to Reviewers is started below. Also, the revised manuscript was again edited for proper English language, grammar, punctuation, spelling, and overall style.

Reviewer 1#

1. The abbreviations should be explained in both abstract and main texts.

R1: We apologize, but we couldn't find any abbreviations in the abstract without definitions. The abbreviations used in the text have been thoroughly examined and adjusted to be defined in the initial citation.

“2. Page numbers are not correct.”

R2: We regret the mistake. The page numbering has been reviewed and corrected according to Reviewer suggestion.

“3. Line 228 is not clear.”

R3: The sentence was revised and corrected.

“4. The order of the figures was suggested to be reorganized based on the results in the texts. For example, in Section 2.1, the figures referred to in the texts followed the order 1, 3, 2…”

R4: The figures were reordered in order to attend the Reviewer suggestion.

“5.  Similarly, in Section 2.2, the Figures referred to followed the order 1, 3, 4, 2... It is hard to find the figure accordingly.” 

R5: We apologize for the mistakes. The figures were reordered in order to attend the Reviewer suggestion.

Reviewer 2 Report

Authors present data from large scale experiment, that could be interesting especially for future breeding programs, as they have found some genotypes, that are performing better under drought stress. Unfortunatelly I am missing some important informations, mainly characterisation of selected genotypes and justification of selection of these particular genotypes (or lineages?). Origin of seeds and at least links to seed/gene bank and/or characterisation of selected genotypes HAS TO BE added into methods (with specification if these are some stable lineages or genotypes still in breeding process etc.), and maybe even into results section - adding an paragraph about genotypes/lineages where they would be introduced to some extend would make it more understandable for the reader, because like this we know nothing about these specific genotypes (e.g. was some selected because it is a regiolal lineage from drier area? does is have better yield in dry season? does it generally have better yield?). Information in lines 411-412 are not enough - based on this statement such data are avaiable, and HAS to be part of the article (or at least references where such data are avaiable, but IMHO it would be better if it would be part of the article).

The article is really descriptive, but rather not conclusive - authors mention biochemical and molecular biological pathways of drought tolerance - is something known about this in best performing genotypes? What is the hypothesis?

Also some future perspectives should be mentioned - will there be further GWAS/TWAS analyses? Will these genotypes be added into some breeding strategies? Are there some markers, that could be used for selection of drought-resistant genotypes?

Why did authors choose 21 days? In my experience (even though with different genera) the early days of germination are critical. Were there some differences in germination - e.g. did all of the genotypes have same germination efficiency or were only those that were able to reach 21 days selected (like from one genotype it was 5 out of 10 pots, for other 9 out of 10 - this would be some selection pressure and would bring bias into the results).

The article is interesting, but without suggested information it is not in state to be published, I believe that after adding required details, it will be usefull to a large scale of readers.

There are some mistakes in word order.

Author Response

We thank the excellent comments of the Reviewers, which we believe that they greatly contributed to improve the quality of manuscript. We have completed all of the changes requested and a long list with changes and responses to Reviewers is started below. Also, the revised manuscript was again edited for proper English language, grammar, punctuation, spelling, and overall style.

Reviewer 2#

“Authors present data from large scale experiment, that could be interesting especially for future breeding programs, as they have found some genotypes, that are performing better under drought stress. Unfortunatelly I am missing some important informations, mainly characterisation of selected genotypes and justification of selection of these particular genotypes (or lineages?). Origin of seeds and at least links to seed/gene bank and/or characterisation of selected genotypes HAS TO BE added into methods (with specification if these are some stable lineages or genotypes still in breeding process etc.), and maybe even into results section - adding an paragraph about genotypes/lineages where they would be introduced to some extend would make it more understandable for the reader, because like this we know nothing about these specific genotypes (e.g. was some selected because it is a regiolal lineage from drier area? does is have better yield in dry season? does it generally have better yield?). Information in lines 411-412 are not enough - based on this statement such data are avaiable, and HAS to be part of the article (or at least references where such data are avaiable, but IMHO it would be better if it would be part of the article).”

R6: We acknowledge the Reviewer comment. In fact, we mistakenly used the term "genotype"; the accurate term should be "cultivar." All soybean and cowpea cultivars used in this study are commercially available and widely disseminated throughout the Brazilian Cerrado region.

Nevertheless, in response to the Reviewer's query, we have made certain amendments to the Material and Methods section and have also created two supplementary tables addressing the cultivar origins, registration data, and other relevant information. Kindly refer to the article's supplementary material for further details. Despite these adjustments, we are certainly open to any additional suggestions.

“The article is really descriptive, but rather not conclusive - authors mention biochemical and molecular biological pathways of drought tolerance - is something known about this in best performing genotypes? What is the hypothesis?”

R7: We do not understand the Reviewer comment. No sentence including molecular and biological pathways of drought tolerance was found in the paper. Also, we did not find these pathways in the literature for all cited cultivars.

In order to attend the Reviewer comment, we add the following hypothesis in introduction:

“Our working hypothesis was that soybean and cowpea cultivars display distinct responses to water deficit, which arise from biochemical adjustments to optimize plant performance. To test this hypothesis, seven semiarid-cultivated soybean and cowpea cultivars were exposed to different water availability levels under greenhouse conditions.”

“Also some future perspectives should be mentioned - will there be further GWAS/TWAS analyses? Will these genotypes be added into some breeding strategies? Are there some markers, that could be used for selection of drought-resistant genotypes?”

R8: In our study, we are currently utilizing released and commercially available cultivars, as mentioned in comment R6. Herein, we aim to disseminate information to farmers, particularly by suggesting the most promising cultivars for cultivation in drought-prone areas. Simultaneously, we are providing insights into defense mechanisms, although limited to biochemical data. While we recognize the importance of breeding studies for ongoing research, it's not our primary focus at the moment.

Moreover, we are conducting in-depth trials involving the pivotal stress-responsive cultivars, integrating metabolome and transcriptome analyses. This data will be used to compose a new paper in the near future. Armed with this supplementary information, we are positioned to make valuable contributions to plant breeding endeavors, aiming to assist plant breeders in developing cultivars even more tolerant to drought conditions.

Why did authors choose 21 days? In my experience (even though with different genera) the early days of germination are critical. Were there some differences in germination - e.g. did all of the genotypes have same germination efficiency or were only those that were able to reach 21 days selected (like from one genotype it was 5 out of 10 pots, for other 9 out of 10 - this would be some selection pressure and would bring bias into the results).

R9: We comprehend the Reviewer query. In this study, the plants were cultivated over a total of 35 days, with 14 days between sowing and the initiation of water stress treatments, followed by an additional 21 days of growth under well-irrigated and water deficit (stress) conditions. During the initial 14 days, seedlings were irrigated in well-watered conditions (maintained at 75% field capacity for both cultivars). At the conclusion of this period, plants were selected for uniformity, and the water stress imposition began. We want comment that all cultivars displayed favorable germination rates, with no discernible impact on subsequent stress treatments.

The duration of 21 days of drought treatments was chosen to align with the conclusion of the vegetative stage. The experiment was monitored daily, revealing that around the midpoint of the third week of water stress, certain plants started flowering (reproductive phase), particularly those subjected to stress, while the well-irrigated plants remained in the vegetative phase. Consequently, to ensure result consistency, we opted to conclude the experiments at the end of the third week (21 days) of water stress exposure.

To enhance clarity regarding the question, we have made adjustments to the Materials and Methods section. Please see the text in red font.

“The article is interesting, but without suggested information it is not in state to be published, I believe that after adding required details, it will be usefull to a large scale of readers.”

R10: We thank the comments. All of Reviewer suggestions were taken into account and incorporated in the text, some of them were justified in Response to Reviewers.

“There are some mistakes in word order.”

R11: We apologize for the errors in the English language. After finishing the suggested changes, the manuscript has been thoroughly revised to ensure that all corrections have been properly addressed, as well as to rectify any potential shortcomings.

Reviewer 3 Report

Comments and Suggestions for Authors

In this article, the authors describe Selection of soybean and cowpea genotypes with superior performance under drought using growth and biochemical aspects.The article is more interesting. However, it needs to be further improved before it could be published.

Specific recommendations:

1. In the conclusion, can the author provide more innovative and constructive ideas to enrich the conclusion with his own thoughts?

2. Please explain the abbreviations that appear for the first time in the article, for example, Line93, explain FC, line94, explain RFM , RDM.

3. Linguistic revisions are needed to improve the readability and clarity of the article.

4. Drought has a large effect on the root system and the authors are advised to design an experiment on the root system. For example, measure the root vigor and do a root analysis.

5. Line 401, Please check that the 2.0℃ symbol is correct.

6. The figure in the article is beautifully done.

7. Line460, please check if the format of 3000x is correct.

8. The authors were asked to analyze the focus of future work based on existing experiments.

Linguistic revisions are needed to improve the readability and clarity of the article.

Author Response

We thank the excellent comments of the Reviewers, which we believe that they greatly contributed to improve the quality of manuscript. We have completed all of the changes requested and a long list with changes and responses to Reviewers is started below. Also, the revised manuscript was again edited for proper English language, grammar, punctuation, spelling, and overall style.

Reviewer 3#

“1. In the conclusion, can the author provide more innovative and constructive ideas to enrich the conclusion with his own thoughts?”

R12: The conclusion was revised and altered in order to attend the Reviewer comment.

“2. Please explain the abbreviations that appear for the first time in the article, for example, Line93, explain FC, line94, explain RFM , RDM.”

R13: All text was completely revised and the definition of abbreviations was included in the first citation.

“3. Linguistic revisions are needed to improve the readability and clarity of the article.”

R14: We sorry for the mistakes. The manuscript was completely revised to ensure that all corrections have been properly addressed, as well as to rectify any potential shortcomings.

“4. Drought has a large effect on the root system and the authors are advised to design an experiment on the root system. For example, measure the root vigor and do a root analysis.”

R15: We appreciate the feedback from Reviewer 3 and understand that a more in-depth analysis of roots would provide additional insights into plant tolerance. However, it's important to highlight that, herein, our primary focus is to provide information about the commercial cultivars that exhibit higher capability for grown under water-limiting conditions.

Furthermore, we want to emphasize that we are currently engaged in numerous plant trials aiming at uncovering cultivation strategies to alleviate drought-induced damages in plants. These strategies include nutrient supplementation, application of crosstalk tolerance inducers, and various other approaches. We will certainly take into account your suggestions to incorporate them into our ongoing studies and conduct assays such as root length, surface area measurement, secondary root count, and others.

Regrettably, we do not have plant material from the current experiment at this time. Nevertheless, we remain completely receptive to any suggestions and feedback that could enhance our work.

“5. Line 401, Please check that the 2.0℃ symbol is correct.”

R16: We sorry for mistake. It was corrected.

“6. The figure in the article is beautifully done.”

R17: We very appreciate the comment. Thank you!

“7. Line460, please check if the format of 3000x is correct.”

R18: The information has been verified and is suitable for utilization in this manner.

“8. The authors were asked to analyze the focus of future work based on existing experiments.”

R19: Good question! Our findings in this current article offer several perspectives. The first one includes the potential for legume producers to gain insights into the available varieties suitable for cultivation in their region, especially when armed with knowledge about the prevailing climatic conditions. Simultaneously, within the scientific research, an opportunity emerges to delve deeper into the biochemical and molecular pathways underpinning the stress tolerance of the presently cultivated cowpea and soybean cultivars, juxtaposed against stress-intolerant counterparts. We are currently directing our efforts along this path.

Furthermore, our data prospect important information for plant breeders to optimize the development of stress-tolerant lineages. We are well aware that this investigation will unlock numerous avenues for actions geared towards cultivating plants under conditions of water scarcity.

“Linguistic revisions are needed to improve the readability and clarity of the article.”

R20: We apologize for the errors in the English language. After finishing the suggested changes, the manuscript has been thoroughly revised to ensure that all corrections have been properly addressed, as well as to rectify any potential shortcomings.

Round 2

Reviewer 2 Report

Authors have adressed all of my points, I hope they agree it helped to improve the paper quality.

Author Response

We are grateful to Reviewer 2 for the comments. Thank you!

Reviewer 3 Report

1.I would suggest that the author make appropriate changes to the language to increase the readability of the article.

2.The authors were asked to analyze the focus of future work based on existing experiments.

I would suggest that the author make appropriate changes to the language to increase the readability of the article.

Author Response

  1. “I would suggest that the author make appropriate changes to the language to increase the readability of the article.”

R2: We would like to express our gratitude to Reviewer 3 for your thorough review of our article. Your insights and suggestions have been incredibly valuable in enhancing the quality of our work. We understand that the Reviewer 3 have suggested certain stylistic changes to the language used in the article, but our team purposely intended to maintain a balance between technical precision and accessibility for a broader readership. We aimed to communicate complex concepts in a manner that both experts and those less familiar with the field can understand.

We recognize that striking this balance can be a challenging task, and we have carefully considered your recommendations. However, we believe that the current style effectively serves our goal of conveying intricate ideas without compromising on accuracy. If you have specific concerns about certain sentences or terminology that you feel could benefit from modification, we would be more than willing to consider those suggestions. Please feel free to point out any instances where the language may impede clarity or precision.

  1. The authors were asked to analyze the focus of future work based on existing experiments

R3: We regret that the previous alterations were not sufficient to address the reviewer's suggestions adequately. In an effort to address your recommendation, we have included additional text in this version, specifically focusing on potential future studies within the discussion section. Furthermore, we have revised the conclusion to enhance its clarity and prominence (see the text in red font). If there are still areas of concern in the text, we kindly request more specific guidance regarding the necessary revisions.